# A Case of Axillary Web Syndrome Caused by Venous Blood Sampling

**DOI:** 10.3390/healthcare11172390

**Published:** 2023-08-25

**Authors:** Hironori Kitajima, Toru Ichiseki, Ayumi Kaneuji, Norio Kawahara

**Affiliations:** Department of Orthopedic Surgery, Kanazawa Medical University, Kahoku 920-0293, Japan; kuppy@kanazawa-med.ac.jp (H.K.); kaneuji@kanazawa-med.ac.jp (A.K.); kawa@kanazawa-med.ac.jp (N.K.)

**Keywords:** axillary web syndrome (AWS), lymphatic vessels, venous blood sampling

## Abstract

Axillary web syndrome (AWS) occurs after breast cancer surgery, sentinel lymph node dissection, or sentinel lymph node biopsy. Here, cord-like structures from the axilla to the forearm limit the range of motion of the shoulder joint and cause pain. Although the etiology is unknown, AWS has been attributed to the blockage of normal lymphatic flow. Here, we report a novel case of AWS after venous blood sampling in a patient. A healthy, 31-year-old male patient experienced pain with a limited range of motion of the shoulder joint the day after venous blood was collected from the left upper extremity for a medical checkup, and he presented to an orthopedic outpatient clinic on the day. Palpation of the axillary region disclosed a cord-like structure in the axillary region of the shoulder joint during abduction, and the patient was diagnosed with AWS. The cord-like structure was noted to be a hypoechogenic luminal structure on ultrasound (US) examination of the axilla, extending from the axilla to below the ulnar cutaneous vein from which the blood was drawn. In patients with pain and a limited range of motion of the shoulder joint, only the shoulder joint is examined during an orthopedic examination. It is important to obtain appropriate physical findings for possible AWS.

## 1. Introduction

Axillary web syndrome (AWS) is a disease that occurs after breast cancer surgery, sentinel lymph node dissection, or sentinel lymph node biopsy in which a cord-like structure extending from the axilla to the forearm limits the range of motion of the shoulder joint and causes pain. AWS was first reported in 2001 by Moskovitz et al. [1]. AWS occurs in between 6 and 91% of cases in different studies depending on several risk factors of postoperative breast cancer patients [2,3] and is considered to be caused by the destruction of lymphatic vessels and veins by axillary surgery and other procedures that block normal lymphatic flow. It has been reported that highly invasive axillary surgery and lean patients are risk factors in the onset of AWS [2].

Although lymphatic involvement has been suggested, the detailed pathophysiology of AWS remains unclear [4,5]. In a study in which lymphoscintigraphy was performed in patients with AWS, it was reported that approximately 90% of the patients had changes such as decreased lymphatic function and obstruction of lymphatic vessels and 67% had collateral lymphatic vessels. These results suggest the involvement of lymphatic vessels in the pathogenesis of AWS [6]. In another report, a biopsy of cord-like structures in seven patients with AWS revealed that lymphatic vessels were involved in three cases, of which a thrombus in the lymphatic vessels was noted in one [7]. A consistent hypothesis is that the cord contains one or more tiny lymphatic vessels tethered to each other or the surrounding tissue by extravasated and cross-linked fluid proteins [8,9]. It is also considered a type of Mondor’s disease in which subcutaneous cord-like induration appears due to inflammation or obstruction of veins running in the shallow subcutaneous layers of the chest, upper abdomen, and upper extremities [6].

The clinical diagnosis of AWS is based on its characteristic physical findings, and imaging tests, such as magnetic resonance imaging (MRI) and ultrasound (US), are considered adjunctive diagnostic tools. However, the sensitivity of MRI in detecting the AWS cord is low. Additionally, it may be challenging to delineate the cord unless the shoulder joint is in abduction [8]. On US examination, findings such as a hypodermic linear structure formed of hyperechogenic walls define a hypoechogenic, hypodermic hyperechogenic linear structure, and hypodermic hypo- or even anechogenic structures with a hyperechogenic intraluminal thrombus. However, no specific findings have been reported [8,9]. Therefore, it is arduous to diagnose AWS using magnetic MRI or US alone.

AWS normally disappears within three months without any therapy [1,2]. However, there are reports that AWS symptoms are still present five years after breast cancer surgery, with extended shoulder joint pain and limited range of motion [3]. Rehabilitation therapy has been proposed for AWS, such as via physical therapies such as soft-tissue mobilization and myofascial release, hot packs, range of motion exercises, strengthening exercises of the shoulder girdle muscles, and manual lymph drainage, but no standardized rehabilitation therapy is currently available [10,11,12].

Venous blood sampling is one of the most common testing procedures performed in hospitals, with over 1 billion procedures performed annually [10]. Complications of venous blood sampling include blood specimens, peripheral neuropathy, complex regional pain syndrome, and vagus nerve reflex [13,14]. In addition, it has also been reported that venous blood sampling from the upper extremity on the affected side after breast cancer surgery may cause lymphedema [15]. Herein, we report a rare case of AWS after venous blood sampling in a patient without a history of axillary surgery. To the best of our knowledge, reports of AWS without direct invasion, such as postoperative breast cancer surgery, sentinel node dissection, or sentinel node biopsy, are rare, with only twelve cases reported in PubMed [16,17,18,19,20,21,22,23,24,25,26,27]. The cases were summarized on 20 May 2023 by searching the keywords ‘Axillary Web Syndrome’ in PubMed. This is the thirteenth case report and the first novel report of AWS occurring after venous blood sampling. In a routine outpatient setting, we report a rare case of AWS as a differential between acute shoulder pain and limited range of motion.

## 2. Case Report

A 31-year-old male patient experienced pain with a limited range of motion of the shoulder joint the day after venous blood was collected from the left upper extremity for a medical checkup, and he presented to an orthopedic outpatient clinic on the day. The medical and family medical histories of the patient were unremarkable. He had no drinking or smoking habits. Pain in the left shoulder joint was elicited by abduction and flexion of the shoulder joint. There were no complaints of nocturnal or night pain. Venous blood was sampled from the ulnar cutaneous vein of the left elbow joint. Blood speculum formation or nerve damage was not observed. The range of motion of the left shoulder joint was 140° in flexion and abduction, with no limitations in internal or external rotation. There was no impingement or painful arc sign in the left shoulder joint. No obvious abnormalities were noted on the radiographic examination of the shoulder joints, such as fractures and osteoarthritis. The patient complained of tightness in the left axillary region. Additionally, when the axillary region was palpated, a cord-like structure was felt in the axillary region during the abduction of the left shoulder joint (Figure 1).

No tenderness or erythema was noted in this cord-like structure, and no skin disease could be noted around the axilla. The cord-like structure was observed as a hypoechoic luminal structure on US examination of the left axilla, extending from the left axilla to just below the ulnar cutaneous vein from which blood was drawn (Figure 2).

There was no blood flow signal in that cord-like structure on Doppler US. Moreover, the US examination did not reveal any deep vein thrombosis or axillary lymphadenopathy in the left upper extremity. Venous blood sampling disclosed no elevation of white blood cells, C-reactive protein, or D-dimer levels. Based on characteristic physical findings, we diagnosed AWS and prescribed analgesics for pain. The patient performed a self-massage on the cord-like structure at home. The cord-like structure gradually disappeared with conservative treatment. Subsequently, the limitation of range of motion improved within three months of diagnosis (Figure 3). The general condition of the patient is stable due to improvement in symptoms without recurrence.

## 3. Discussion

In the present case, an axillary cord-like structure after venous blood sampling was noted in a healthy adult patient. Ultrasonographic findings revealed cord-like structures extending from the axillary region to just below the ulnar cutaneous vein, where venous blood sampling was performed. Therefore, a minor injury to the lymphatic vessels running parallel to the ulnar vein on venous blood sampling may have contributed to the onset of AWS. Although it is expected that symptoms would occur distal to the site of injury based on the lymphatic flow, in this case, AWS occurred proximal to the site of the injury. We hypothesized that AWS was caused by impaired lymphatic function in injured lymphatic vessels triggered by venous blood sampling and abnormal lymphatic return due to microthromboemboli, which was undetected by the US.

Twelve cases of AWS other than breast cancer surgery, sentinel node dissection, or sentinel node biopsy have been reported (Table 1). The age of onset ranges from 27 to 73 years, and it occurs in both males and females. The onset of AWS symptoms ranged from 24 h to 17 days after the trigger. As for AWS’s etiology, three cases were idiopathic AWS with no known cause [18,20,26]. Moreover, four cases were reported as possibly induced after physical activity or trauma [17,21,25,27], and the possibility of lymphatic involvement due to physical activity or trauma was mentioned. The physical activities were squash sports, jumping rope, and supporting one’s weight with one’s arms. The trauma was caused by tripping on the sidewalk and bumping the elbow, i.e., a minor trauma without fracture. It is suggested that AWS may be induced even by such minor trauma or physical activity. Next, three cases of AWS due to skin disease in the axillary region [16,19,24] were reported. These reports include folliculitis, furuncles, and epidermal inclusion cysts, and the involvement of axillary inflammation associated with skin disease in the pathogenesis of AWS has been discussed. In all cases of AWS associated with axillary skin disease, treatment of the skin disease was practiced, and the possibility that treatment of the cutaneous disease (which appears to be the primary cause of AWS) may help treat AWS has been described. Other reports of AWS with collagenase injections into the palms performed for Depuytran’s contracture have also been reported [22]. It has been reported that lymphadenopathy in the axillary region occurs as a side effect of collagenase injections into the palms [28], and the involvement of the lymphatic system affected by collagenase injections has been described. Finally, one case of AWS has been reported due to lymphatic vascular damage associated with axillary lymph node dissemination of tuberculosis [23]. In this report, drug-induced tuberculosis was treated. This is the thirteenth case reported and the first case reported in the literature concerning an outbreak after venous blood sampling. In 10 of these cases, including the present case and excluding idiopathic cases, some lymphatic involvement was suspected as a cause of AWS. In general, AWS usually improves after about three months of conservative treatment. In twelve reported cases, the symptoms improved in all cases within a few weeks to 3 months. One patient was reported to have residual cord-like structures without residual symptoms [22]. Therefore, these reports of AWS suggest that AWS, other than axillary surgery, can be treated conservatively and improved in the same way as usual AWS. However, if there is an axillary skin disease or a disease that causes axillary lymphadenopathy, which may be the underlying cause of AWS, the underlying disease should be treated.

In recent years, physiotherapy has received increasing attention as a treatment for AWS after breast cancer surgery [10,11,12]. Since it has been reported that the presence of AWS after breast cancer surgery results in decreased shoulder joint range of motion five years later [3], an appropriate physical therapy protocol for AWS is important for maintaining activities of daily living (ADL). In addition, persistent shoulder joint pain and limited range of motion may induce atrophic muscle weakness in the shoulder joint. Currently, various physical therapies such as soft-tissue mobilization and myofascial release, hot packs, range of motion exercises, strengthening exercises of the shoulder girdle muscles, and manual lymph drainage have been indicated for the treatment of AWS [10,11,12]. However, no standard physical therapy protocol has been established. Borg et al. reported that a physiotherapy protocol combining manual lymphatic drainage, manual therapies (including soft tissue, cording, and scar mobilization), shoulder stretching, and mobility exercises improved range of motion limitation and decreased pain in 98% of AWS cases. This physical therapy protocol also resulted in no adverse events [10]. Torres-Lacomba et al. compared physiotherapy with and without manual lymphatic drainage in an RCT and found that manual lymphatic drainage helped reduce pain [5]. It is expected that physiotherapy protocols for AWS after breast cancer surgery will become standardized in the future. In addition, whether physiotherapy is helpful for AWS other than axillary surgery needs to be studied. To date, physiotherapy intervention has been used in five cases of AWS other than axillary surgery [17,18,19,22,26], and all have had a suitable course of treatment. These included soft-tissue mobilization and myofascial release, hot packs, range of motion exercises, strengthening exercises of the shoulder girdle muscles, and manual lymph drainage. It is suggested that physical therapy may help prevent residual range of motion limitations in AWS other than axillary surgery.

A clinical algorithm regarding the diagnostic and therapy of AWS is presented (Figure 4). Listing AWS as a differential condition between acute shoulder pain and limited range of motion is challenging. However, AWS can be caused by minor lymphatic injuries due to venous blood sampling or even minor trauma or physical activity, so the prevalence of AWS is likely to be higher than it should be. Clinicians need to recognize AWS because its characteristic physical findings directly relate to the diagnosis. It is also important to understand that AWS can occur other than after axillary surgery and to take appropriate physical examination findings. Due to its low recognition, the diagnosis may be missed and misdiagnosed as unexplained shoulder pain, such as periarthritis or frozen shoulder. Appropriate diagnosis of AWS will reduce unnecessary medical costs due to excessive testing and provide adequate informed consent to the patient. In addition, physical therapy treatment options may be offered to improve shoulder pain and range of motion and prevent prolongation of shoulder pain.

## 4. Conclusions

We report a rare case of AWS caused by a minor lymphatic injury during venous blood sampling. Limited range of motion and pain in the shoulder joint with axillary tightness should be considered as possible AWS. AWS is diagnosed based on characteristic physical findings, so it is important to obtain appropriate physical findings. If a diagnosis of AWS can be made, observation can lead to improvement without treatment, usually within three months. Optional physical therapy may enhance treatment.

## Figures and Tables

**Figure 1 healthcare-11-02390-f001:**
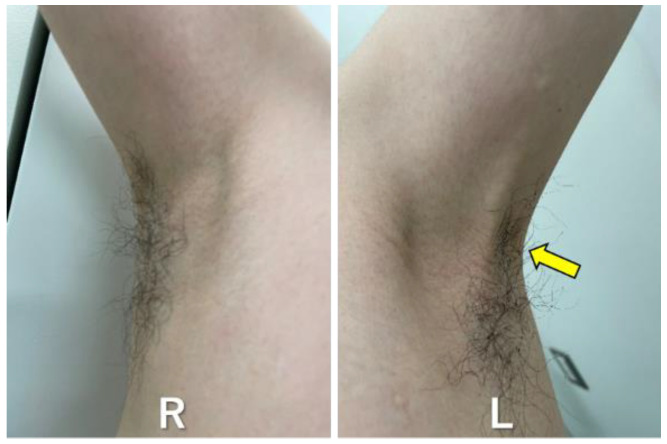
Bilateral axillae at the time of examination. A cord-like structure is observed in the left axilla (yellow arrow). R indicates right axillae, and L indicates left axillae.

**Figure 2 healthcare-11-02390-f002:**
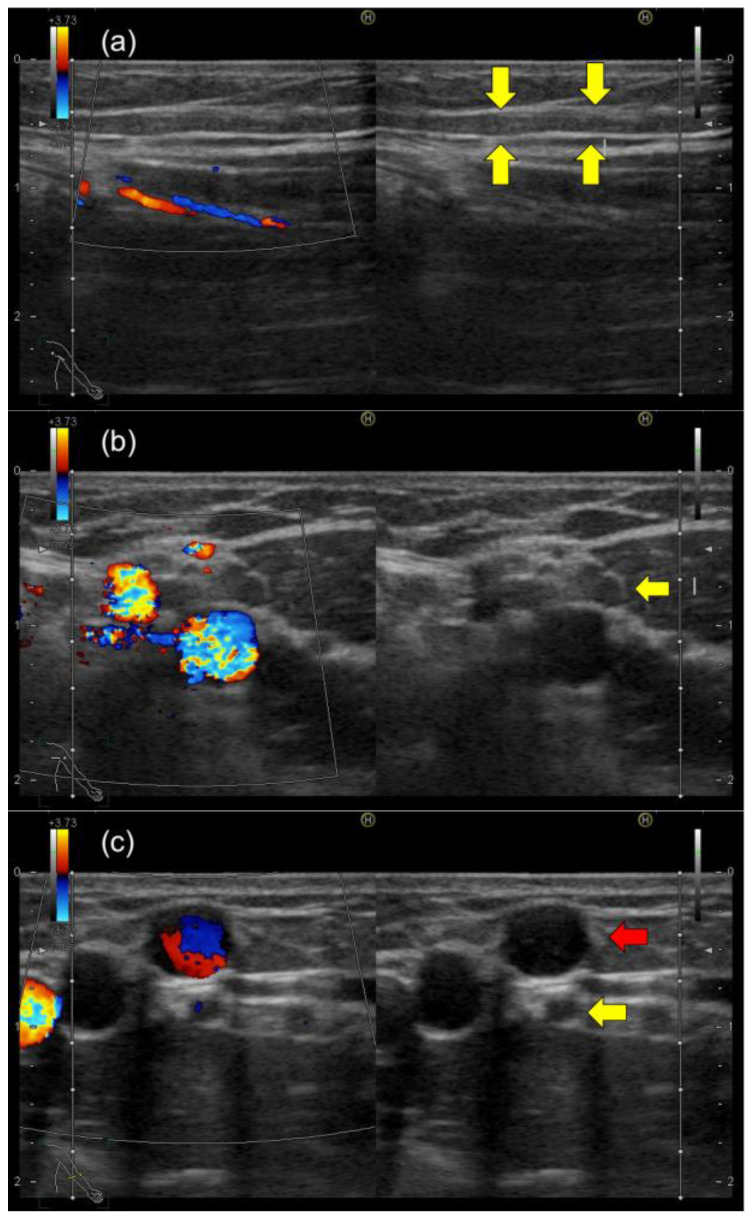
Ultrasound image of the left axillary region during an outpatient visit: (**a**) sagittal ultrasound image of the axilla; (**b**) axial ultrasound image of the axilla (hypoechogenic luminal structures and hyperechogenic walls are seen (yellow arrow)); (**c**) ultrasound image of the left elbow, where venous blood sampling was performed. The cord-like structure (yellow arrow) is observed as a hypoechoic luminal structure on US examination of the left axilla, extending from the left axilla to just below the ulnar cutaneous vein (red arrow) from which the blood was drawn.

**Figure 3 healthcare-11-02390-f003:**
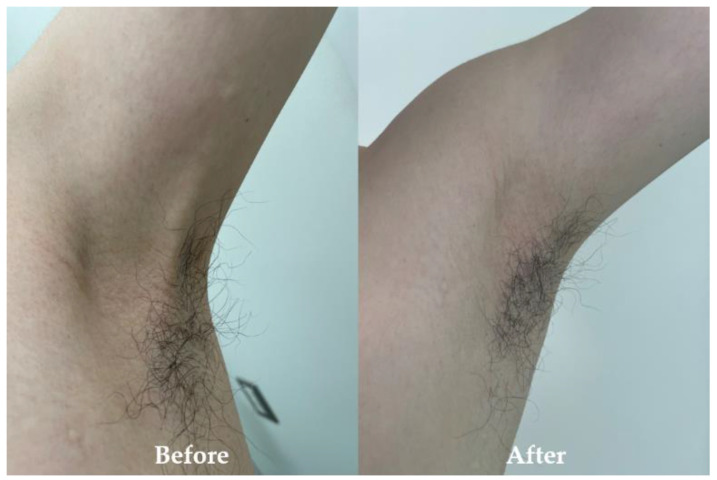
Left axilla at the first visit (Before) and left axillary region three months after onset (After). The cord-like structures in the left axilla have disappeared.

**Figure 4 healthcare-11-02390-f004:**
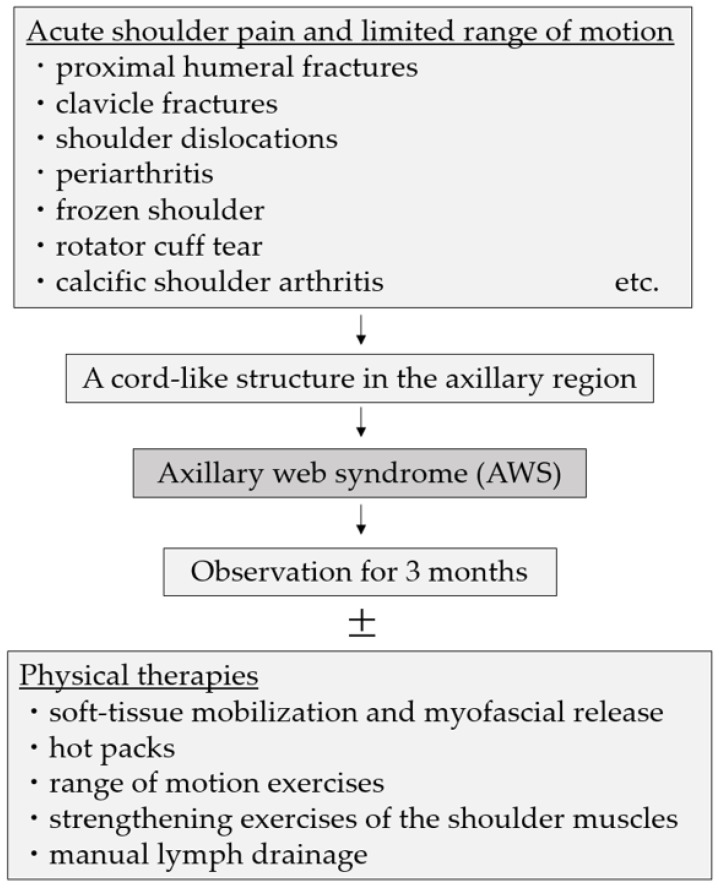
A clinical algorithm regarding the diagnostic and therapy of AWS.

**Table 1 healthcare-11-02390-t001:** Twelve cases of AWS other than breast cancer surgery, sentinel node dissection, or sentinel node biopsy.

Case	Year	Age	Sex	Origin	Time to Onset	Rehabilitation	Time to Improvement	Additional Treatment
Rashtak et al. [24]	2012	57	M	Furuncle	unknown	−	5 weeks	Antimicrobial therapy for furuncle
Zhang et al. [16]	2016	27	F	Granulomatous Inflammation after Folliculitis	unknown	−	3 months	Excision of the nodule
Welsh et al. [17]	2016	38	M	Squash (sport)	24 h	+	1 months	−
Demir et al. [18]	2017	40	M	Idiopathic	unknown	+	1 months	−
Lee et al. [19]	2019	63	M	Epidermal inclusion cysts	unknown	+	2 months	Excision of the nodule
Puentes et al. [20]	2020	67	F	Idiopathic	unknown	−	4 months	−
Hunt et al. [21]	2020	43	M	Supporting one’s weight with one’s arms	5 days	−	a few weeks	−
Soares et al. [22]	2021	65	F	Collagenase injection for Dupuytren’s contracture	17 days	+	2 months	−
Malek et al. [23]	2021	42	M	Cutaneous Tuberculosis and Lymphatic Dissemination	7 weeks	−	3 months	Chemotherapy for Tuberculosis
Alharazy et al. [25]	2023	38	M	Jumping rope	2 days	−	8 weeks	−
Siddiqui et al. [26]	2023	73	F	Idiopathic	unknown	+	1 week	−
Mohammed et al. [27]	2023	63	M	Trauma	2 weeks	−	2 months	−
This report	2023	31	M	Venous Blood Sampling	24 h	−	3 months	−

## Data Availability

Not applicable.

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
