# Peer review of "A Case of Axillary Web Syndrome Caused by Venous Blood Sampling"

_healthcare, 2023, doi:10.3390/healthcare11172390_

Round 1

Reviewer 1 Report

This is my review report for manuscript: A Case of Axillary Web Syndrome Caused by Venous Blood Sampling

The article is about case report od AWS after routine venous blood sampling, which makes this article more interesting for potential readers although AWS is not such a rare occurrence and appears in a significant number of patients after axillary surgery

The article is adequately designed, it has main necessary elements for case report, but need minor revision about bellow proposed questions.

Please write correctly: sentinel lymph node biopsy, not sentinel node dissection. You repeat the same phrase several times throughout the text, it might be more appropriate to write only axillary surgery

How much time past from the blood sampling until the onset of shoulder pain, i.e. until seeking medical examination for the mentioned symptoms? Please write this in case presentation

Have you consulted a specialist of physical medicine and rehabilitation regarding potential physical therapy?

Was a follow-up ultrasound of the left axilla performed?

English language require minor editing.

English language require minor editing.

Reviewer 2 Report

Dear authors,

I’ve reviewed the manuscript "A Case of Axillary Web Syndrome Caused by Venous Blood Sampling".

I want to congratulate the authors for providing a highly relevant and moreover, clinically useful case report. However, I would like to share my points of critique/suggestions with the authors.

- no "CARE Case Report Guidelines" were used (https://www.care-statement.org/). This is a missing quality aspect in this study. Apply these criteria and revise the whole manuscript according to CARE guidelines.

Abstract

-       Clarify „US“ – please spell in full this abbreviation.

-       Clarify “subsequently” (exact time point)

Introduction

-       Please delete the first part of the introduction (lines 1-15). Here, the authors only describe shoulder pain without giving information regarding AWS.

-       Add more information regarding AWS and pathophysiological / clinical / epidemiological data

-       Lines 29-30: reference is missing (“Additionally, it may…”)

-       Clarify the abbreviation “US”

-       The authors wrote, “The treatment of AWS often shows improvement after about three months of conservative treatment.” The authors should mention that this disease normally disappears within three months without any therapy (incl. reference)

-       “Rehabilitation therapy has been proposed…” – please mention and describe physical therapy which is commonly used in conservative treatment of AWS (incl. reference)

-       “…only twelve cases reported in PubMed” – add access date of the search and which search terms / search strategy was used?

Case report:

-       Please add exact time point of blood sampling; how many days / weeks / months did it take, that the patient got symptoms and presented to the orthopaedic outpatient clinic?

-       Why was blood sampled from this patient?

Figures:

-       Figure 1: Description of “R” and “L” is missing

-       Figure 3: Please add before picture, and merge before and after pictures.

Discussion:

-       Lines 1-3: reference is missing

-       Clarify the abbreviation “ADL”

-       “In addition, the pain and limited range of motion associated with AWS may induce periarthritis…”: reference is missing.

-       “Maria Torres-Lacomba et al.” – please delete “Maria”

-       Add an illustrative (clinical) algorithm regarding diagnostic and therapy of AWS

-       Twelve similar cases are reported in PubMed. Please summarize these cases in more detail (e.g. time of symptom presentation, therapy yes / no, which therapy and how long, age, gender distribution). To obtain a good overview, the most important clinical aspects of these cases should be presented in a table. Highlight similarities and differences of these cases, and compare them with the reported case in this report.

Conclusion:

-       Delete the sentence “When a patient presents …” – this is a confusing sentence without a message

-       Enhance this section with clinical suggestions regarding symptoms and therapy and high-risk patients (e.g. symptoms, when do the symptoms occur, diagnostic and therapy modalities).

- What is the message of this report to clinicians? 

English language:

The whole article needs to be extensively edited for the English language. There are some spell errors, grammar, and typing errors. There are some spell errors, grammar, and typing errors.

The whole article needs to be extensively edited for the English language. 

Round 2

Reviewer 2 Report

Dear authors,

thank you for implementing my suggestions and comments. After the revision, the manuscript significantly improved.

I congratulate the authors. 

Best regards

Moderate editing of English language required.